# SVD-DIP: Overcoming the Overfitting Problem in DIP-based CT Reconstruction

**Marco Nittscher**[*]                                      MNITTSCH@UNI-BREMEN.DE
*Center for Industrial Mathematics, University of Bremen, Germany*

**Michael Lameter**[*]                                      LAMETER@UNI-BREMEN.DE
*Center for Industrial Mathematics, University of Bremen, Germany*

**Riccardo Barbano**                              RICCARDO.BARBANO.19@UCL.AC.UK
*Department of Computer Science, University College London, UK*

**Johannes Leuschner**                                      JLEUSCHN@UNI-BREMEN.DE
*Center for Industrial Mathematics, University of Bremen, Germany*

**Bangti Jin**                                                  B.JIN@CUHK.EDU.HK
*Department of Mathematics, The Chinese University of Hong Kong, Shatin, N.T., Hong Kong*

**Peter Maass**                                              PMAASS@UNI-BREMEN.DE
*Center for Industrial Mathematics, University of Bremen, Germany*

**Editors:** Accepted for publication at MIDL 2023

## Abstract

The deep image prior (DIP) is a well-established unsupervised deep learning method for image reconstruction; yet it is far from being *flawless*. The *DIP overfits to noise* if not early stopped, or optimized via a regularized objective. We build on the regularized fine-tuning of a pretrained DIP, by adopting a novel strategy that restricts the learning to the adaptation of *singular values*. The proposed SVD-DIP uses *ad hoc* convolutional layers whose pretrained parameters are decomposed via the singular value decomposition. Optimizing the DIP then solely consists in the fine-tuning of the singular values, while keeping the left and right singular vectors fixed. We thoroughly validate the proposed method on real-measured µCT data of a lotus root as well as two medical datasets (LoDoPaB and Mayo). We report significantly improved stability of the DIP optimization, by *overcoming the overfitting* to noise.

**Keywords:** Deep Image Prior, Fine-Tuning, Computed Tomography, Singular Value Decomposition

## 1. Introduction

In medical imaging, we are often interested in inverse problems of the form $y = Ax + \nu$, with $y \in \mathbb{R}^m$ being a noisy measurement, $x \in \mathbb{R}^n$ the unknown image of interest, $A$ a linear forward operator, and $\nu \sim \mathcal{N}(0, \sigma^2 I)$ i.i.d. noise. This problem is often ill-posed, and regularization is needed to recover a sensible image (Engl et al., 1996; Ito and Jin, 2015).

In recent years, deep learning has been applied successfully to many imaging modalities, often via a supervised learning paradigm (Arridge et al., 2019; Ongie et al., 2020). However, supervised learning tends to require a large amount of paired training data to be effective (Baguer et al., 2020). Deep Image Prior (DIP) (Ulyanov et al., 2020) is an unsupervised

---

[*] Contributed equally

alternative to applying deep learning to image reconstruction. Its main advantage over supervised methods is that it requires no training data, and learns only on the observed data sample, relying on the rich structure of convolutional neural networks (CNNs) to have a regularizing effect on the image. However, it is not without limitations. The network can overfit to the noise, and has to be freshly trained for every image we intend to reconstruct. The Educated Deep Image Prior (EDIP) (Barbano et al., 2022) is a variant of DIP that addresses some of these issues. It uses pretraining by initializing the network architecture with a pretrained (warm-start) parameter setting, rather than randomly. However, EDIP still suffers from overfitting even if equipped with a regularized objective; iterating beyond a certain point leads to deteriorated quality.

The problem of overfitting is not unique to DIP. Recently, Sun et al. (2022) suggested a novel way of adapting pretrained parameters in a CNN using the singular value decomposition (SVD) to address overfitting and the generalization ability of a CNN specifically in the context of image segmentation. In this paper, our contribution is to integrate the idea of the SVD fine-tuning into the EDIP framework, achieving remarkable stability. Specifically, we first train a CNN on synthetic data, and before warm-starting the DIP with the pretrained parameter setting, we replace some or all of the layers of the network using the SVD. In our experiments, the resulting DIP (i.e., SVD-DIP) is much more resistant to overfitting. When iterating for very long, e.g. 200k iterations, it retains a high PSNR value, outperforming the classical DIP. On the other hand, the SVD-DIP only leads to a minor drop in the maximal PSNR value.

## 2. The Deep Image Prior

Given a measurement $y$, the DIP (Ulyanov et al., 2020) finds an $x$ that minimizes $||Ax-y||_2^2$. The method trains a CNN, typically deploying a U-Net architecture (Ronneberger et al., 2015), to fit to the single data sample $(z, y)$, where $z$ is a randomly initialized input (e.g., with i.i.d. Gaussian noise). The DIP finds a parameter setting $\theta$ such that the output of the neural network $\varphi_\theta(z) = x$ minimizes the error $||A\varphi_\theta(z) - y||_2^2$. This requires training a CNN for each $y$; the optimization can take many hours, depending on the complexity (esp. high-dimensionality) of the reconstruction task (Barbano et al., 2022). The method relies on the observation that the CNN structure already captures a sufficient amount of low-level image statistics, such that it can reconstruct an image well even without being trained on any data save for the input image (Dittmer et al., 2020). When data is scarce or expensive to acquire, this represents a major upside. However, the Achilles' heel of the DIP is overfitting to noise. When optimizing for too long, the network fits the noise. There are several approaches to alleviate this issue. The use of an explicit regularization term (Liu et al., 2018) is one of them. For all our DIP variants, we replace the standard objective used in Ulyanov et al. (2020) $||A \cdot -y||_2^2$ with

$$||A \cdot -y||_2^2 + \gamma \text{TV}(\cdot) \tag{1}$$

where $\gamma > 0$ and $\text{TV}(x) := \sum_{i,j} |x_{i+1,j} - x_{i,j}| + \sum_{i,j} |x_{i,j+1} - x_{i,j}|$ is the anisotropic total variation. That is, the DIP baseline uses the regularized objective in Equation 1. The latter only partially alleviates overfitting to the noise, often not being sufficient to prevent it completely. Other methods rely on the Stein's unbiased risk estimator (Jo et al., 2021)

(also see Appendix F), or on hand-crafted early-stopping criteria (Liu et al., 2018; Wang et al., 2021). Additionally, one may constrain the network to an under-parameterized regime such that its ability to fit to noise is reduced, e.g., deep decoder (Heckel and Hand, 2019).

The Educated Deep Image Prior (EDIP) (Barbano et al., 2022) is a variant of the DIP that makes use of pretraining, motivated by wanting to speed up the DIP, since freshly training DIP on multiple measurements is time-consuming. The idea is to first train the CNN architecture using available or synthetic data. The obtained parameter setting $\theta^\star$ is used as the starting point for the subsequent DIP reconstructive task on $y$. This "warm-start" allows the network to reconstruct the desired image in fewer iterations, exploiting benign inductive biases learned in the pretraining phase.

## 3. Overcoming Overfitting via an SVD-based Pretrained CNN

While the original EDIP model was motivated by a desire to speed up the DIP, we are instead interested in utilizing pretraining to address overfitting. Intuitively, we can think of the warm-start parameter setting obtained via pretraining as a list of tensors $W$, each coding the weights for a convolutional layer in the CNN. In EDIP, this parameter setting is used as the initialization. Thus, the only mechanism in place to stop from eventually overfitting to noise is the TV term in the loss. Our approach, the SVD-DIP, integrates the idea from (Sun et al., 2022) of adapting the pretrained parameter setting, into the DIP framework. Specifically, the pretrained tensor $W \in \mathbb{R}^{C_{out} \times C_{in} \times K \times K}$ (which represents $K \times K$ convolutions) is first folded into a matrix $W' \in \mathbb{R}^{C_{out} \times C_{in}K^2}$. Then, the SVD yields three matrices, $U' \in \mathbb{R}^{C_{out} \times R}$, $S' \in \mathbb{R}^{R \times R}$ (diagonal), $V' \in \mathbb{R}^{R \times C_{in}K^2}$ $\quad (R = \min\{C_{out}, C_{in}K^2\})$,

$$W' = U'S'V'.$$

These matrices can be "unfolded" back into three tensors $U \in \mathbb{R}^{C_{out} \times R \times 1 \times 1}$, $S \in \mathbb{R}^{R \times R \times 1 \times 1}$, $V \in \mathbb{R}^{R \times C_{in} \times K \times K}$ such that composing the tensors' induced convolutions is equivalent to taking the original tensor's induced convolution. For the technical details, see Appendix A. The key idea of the proposed approach is to represent the network parameters in this form, then freeze the tensors $U$ and $V$, keeping their convolutions fixed, but allowing the tuning of the singular values (SVs), i.e. the diagonal of $S'$.

Freezing $U, V$ and only varying the SVs retains the structure learned by pretraining the network, but the weighting of these structural components can be fine-tuned for a specific task at hand: increasing the weight of more important structures, or reducing less important ones. Figure 2 shows that the SVs of the pretrained CNN are modified by SVD-DIP, but still retain the overall trend of the original values.

Prior to the DIP fine-tuning, the network encodes the same function whether it is warm-started with the pretrained parameter setting, or its SVD-adapted counterpart, since with or without the SVD replacement, each layer performs exactly the same convolutional operation. Once fine-tuning begins, the two methods diverge from each other with SVD-DIP being more constrained due to having fewer parameters. The way the SVD is used to replace convolutional layers can be viewed as parameter compression; if $C_{in} = C_{out}$, the number of SVs is a fraction of the square root of the number of parameters of the original tensor. This drastic reduction in the number of parameters, while still retaining enough expressive power thanks to the structure learned during pretraining being encoded in $U$ and

$V$, greatly alleviates the overfitting issue and stabilizes the convergence, as demonstrated by the experiments below. When constraining the parameter space, we sacrifice some capability to represent finer structure, but gain substantial robustness to overfitting. We are more interested in a proof of concept for a method with high stability, which bypasses the need for reliable early stopping criteria, even when the pretraining and application datasets are different.

## 4. Experiments and Results

### 4.1. The Experiment Setup

We conduct our experiment on both µCT and medical CT image datasets. During pretraining, the U-Net learns to post-process filtered back-projections of noisy CT projections, that are simulated using the geometry of the target data. See Appendix B for more details. We consider the following scenarios:

**Pretraining on Ellipses → fine-tuning on Lotus and LoDoPaB's Chest** Pretraining is performed on a dataset consisting of synthetically generated ellipses, which is commonly used for inverse problems arising in imaging (Adler and Öktem, 2017) and used to warm-start DIP for CT reconstruction in (Barbano et al., 2022). Synthetic data is advantageous in applications where data is scarce. 5% Gaussian noise is added to the simulated projection data. We use this approach for two target datasets: (i) real-measured µCT data of a lotus root (Bubba et al., 2016) (fan-beam geometry with 20 angles, 429 detector pixels and image size $128 \times 128\,\text{px}^2$); (ii) simulated medical chest CT data from LoDoPaB (Leuschner et al., 2021) (parallel-beam geometry with 200 angles, 513 detector pixels and image size $362 \times 362\,\text{px}^2$). It is simulated to include Poisson noise corresponding to 4096 photons per pixel before attenuation. As per best practice, the ground truth images with low artifact corruption were preferred when selecting 10 LoDoPaB test samples via manual inspection before running any experiments. Note that there is a shift in noise distribution between pretraining that uses 5% Gaussian noise and the target data which is either real-measured or simulated to contain Poisson noise. The U-Net architecture is shown in

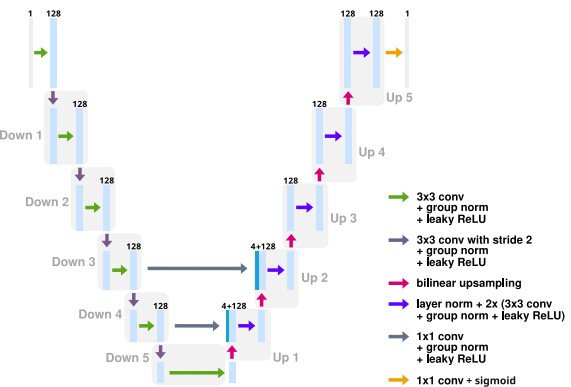

Figure 1: U-Net architecture used for DIP, as well as for EDIP and SVD-DIP when pretraining on Ellipses.

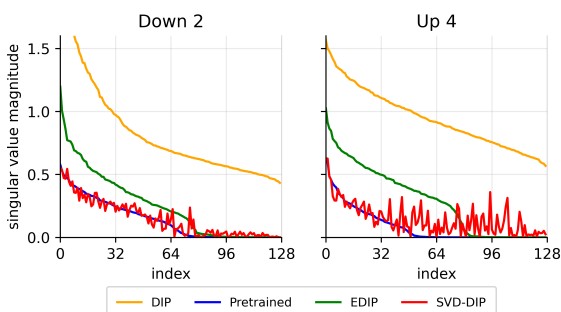

Figure 2: SVs taken from the 1st convolutions of Down layer 2 and Up layer 4 of the U-Net on the lotus.

Figure 1, based on (Baguer et al., 2020, Table B5, DIP for LoDoPaB). The SVD replacements reduce the number of parameters from $128^2 \cdot 3^2$ to 128 in each convolution layer, i.e., reducing by a factor of $\frac{1}{1152}$. We refer to the two transfer settings as "Ellipses-Lotus" and "Ellipses-LoDoPaB".

**Pretraining on Chest → fine-tuning on Mayo's Head, Abdomen and Chest** Pretraining is performed on LoDoPaB (Leuschner et al., 2021). The dataset contains ca. 40k chest CT images of size $362 \times 362 \, \mathrm{px}^2$ and simulated parallel-beam projections with 1000 angles, 513 detector pixels and Poisson noise corresponding to 4096 photons per pixel before attenuation. We experiment with both the original 1000-angle geometry as well as a sub-sampled 200-angle geometry. We use images of different body parts from (Moen et al., 2021) as the images to be reconstructed and simulate observations using the same geometry and noise setting that was used in LoDoPaB. For EDIP and SVD-DIP, we use the U-Net architecture shown in Appendix C Figure 8(b) trained in (Baguer et al., 2020), while for DIP we use the architecture shown in Figure 1, which is optimal in non-pretrained settings (see Appendix C). We refer to this setting as "LoDoPaB-Mayo".

We use the resulting parameters as our pretrained parameters. Both EDIP and SVD-DIP receive the filtered back-projection obtained from test data, $z = \mathrm{FBP}(y)$, matching the post-processing task of the pretraining, while DIP receives an i.i.d. Gaussian noise image $z$. For the target data containing Poisson noise (Ellipses-LoDoPaB and LoDoPaB-Mayo), we replace the squared error $\|A \cdot - y\|_2^2$ in the objective (1) with the Poisson regression loss matching the noise distribution. A suitable regularization parameter $\gamma$ is chosen for each setting, and the same value is used for all methods. See Appendix B for more details. We repeat three times for each image from the data sets, for each variant of DIP. We test the classical (randomly initialized) DIP, EDIP and SVD-DIP.

## 4.2. Comparison of Methods

### 4.2.1. μCT of lotus root (Ellipses-Lotus)

Figure 3 shows the convergence behavior of the different DIP frameworks for the lotus. EDIP peaks early, reaching a maximum PSNR (max PSNR) value of 31.95 dB. However, after the peak, its PSNR steadily decreases, indicating overfitting. The vanilla DIP and the (pretrained) SVD-DIP are on par in terms of the max PSNR (31.69 dB and 31.60 dB, resp.), but the vanilla DIP is far less stable, i.e. its PSNR falls after the peak value, while the PSNR of SVD-DIP does not decline. While EDIP shows a similar overfitting behavior as DIP, EDIP reports a higher max PSNR. If we consider both max PSNR and overfitting, pretraining generally improves the DIP, with EDIP achieving the best max PSNR and SVD-DIP avoiding overfitting. Thus, the pretrained SVD-DIP performs almost as well as EDIP in terms of max PSNR and clearly outperforms EDIP in terms of stability. EDIP has the benefit of the speed at which it reaches its best PSNR value.

As an ablation study, we also include the results for SVD-DIP without pretraining, which performs much worse than the rest. This is not surprising, since it can only learn a weighting of the singular vectors that posses no useful structure, due to the random initialization.

**SVD Truncation**  Since before and after the fine-tuning, a large portion of SVs close to zero generally remain so, we observe that the SVD-DIP can be further reduced in the number of parameters by the truncated SVD. Figure 4 contains a comparison of SVD-DIP settings with different approaches to truncation. For (50% rd.), we truncated all but the top 50% of the SVs. For (10% thresh.), we truncated all SVs which are below 10% of the largest SV. The graph shows that the different variations of SVD-DIP perform very similarly, the difference between their PSNRs being within the order of magnitude of the standard deviation observed for repeated runs of the same experiments. Thus, the lotus root image is simple enough to allow for a low-dimensional representation of the parameter space by only the upper 50% of the SVs (50% rd.), or only the values greater or equal to 10% of the largest (10% thresh.). We view these results as a proof-of-concept for SVD truncation, whose advantageous utilization would be subject of future work.

### 4.2.2. Chest CT (Ellipses-LoDoPaB)

Table 1: Mean PSNR values for Ellipses-LoDoPaB (Sparse 200) over 3 runs and 10 images, 50000 Iterations each. Max denotes the maximum PSNR, Final denotes the PSNR attained after 50000 iterations. Values are in dB. Table 4 in the Appendix contains the data for all runs on which we compute the mean.

| | DIP | | Pretraining | EDIP | | SVD-DIP | |
| | Final | Max | Init | Final | Max | Final | Max |
|---|---|---|---|---|---|---|---|
| Mean | 32.08 | 35.34 | 30.51 | 32.39 | 35.09 | 34.65 | 34.76 |

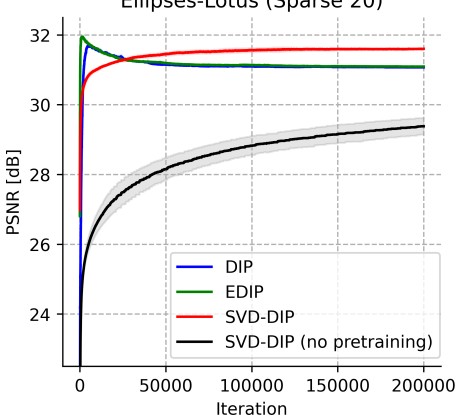
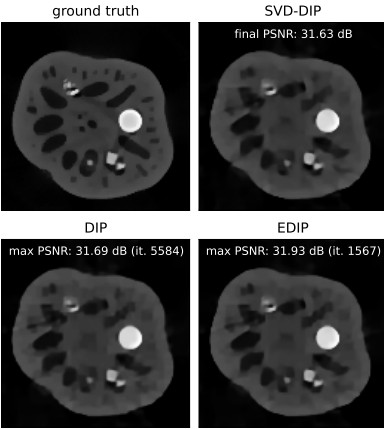

Figure 3: Optimization of DIP, EDIP and SVD-DIP on the lotus, pretraining on ellipses. Each line represents the mean over 3 runs, the colored area the standard deviation. The final SVD-DIP reconstruction is shown on the right, along with the max PSNR reconstructions of DIP and EDIP, which would require ideal early stopping.

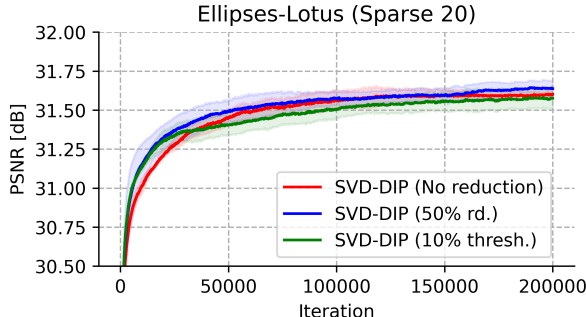

Figure 4: Optimization of SVD-DIP for forms of truncation on the lotus, pretraining on ellipses. Each line represents the mean over 3 runs, the colored area the SD.

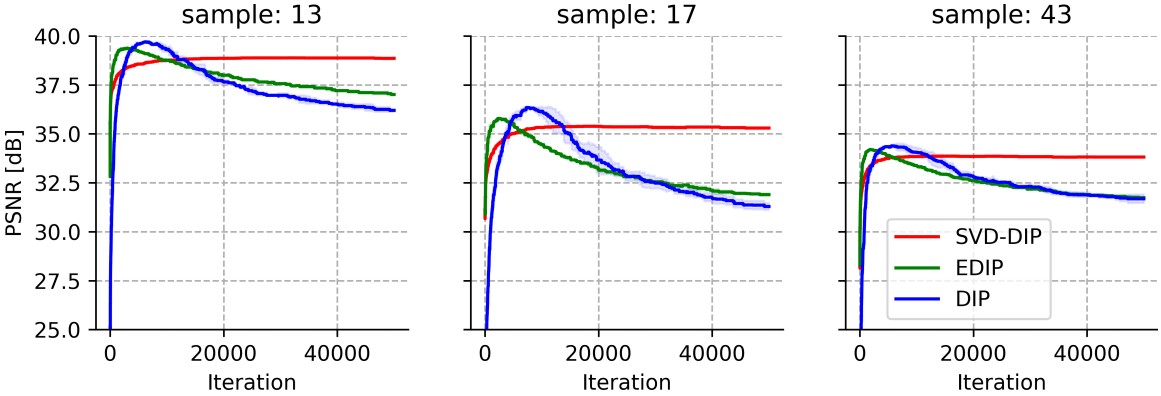

Figure 5: Optimization of DIP, EDIP and SVD-DIP on LoDoPaB (Sparse 200) test samples 13, 17 and 43, pretraining on ellipses. Each line represents the mean over 3 runs.

DIP has a better max PSNR on average (see Table 1). SVD-DIP reaches on average a lower max PSNR than EDIP and DIP, but it is much more stable, with the PSNR trending upwards, rather than declining. Figure 5 exemplarily shows the behavior of the PSNR observed in the runs. On average, SVD-DIP outperforms DIP and EDIP, arriving at a slightly lower value in a remarkably stable manner. Thus, the strong regularization induced by fixing the singular vectors comes at the cost of a slightly lower max PSNR, but on the other hand greatly reduces the overfitting. Here, for SVD-DIP we replace the convolutional layers in all down- and up-blocks by SVD representations, except the first three down-blocks. We can treat the number of down-blocks that remain unchanged as a hyperparameter, which can be adjusted depending on the need to adapt to different input data. We replace fewer blocks as we have a major change in the noise distribution, so as to provide the network with greater flexibility.

### 4.2.3. CT of different body parts (LoDoPaB-Mayo)

As shown in Figure 6, SVD-DIP on average reports higher final PSNR values when compared to EDIP and DIP, as well as max PSNR values similar to those of EDIP. On Chest and Abdomen, max PSNR values are comparable across all methods; on Head instead, the

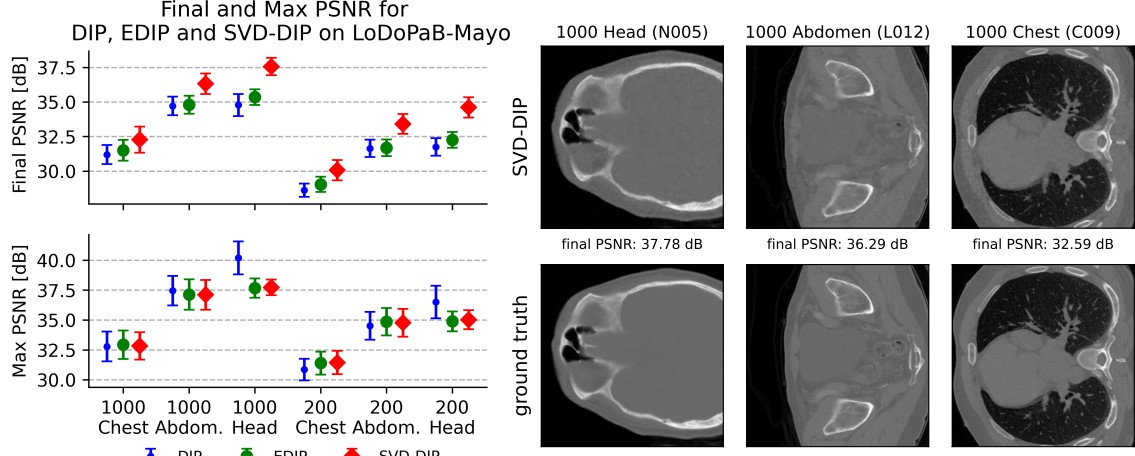

Figure 6: The mean and SD of Final and Maximum PSNR for DIP, EDIP and SVD-DIP on different body parts from LoDoPaB-Mayo, pretraining on LoDoPaB. Both 1000 and 200 angle settings are considered, indicated on the horizontal axis. For each setting, 10 samples with 3 runs each are considered. Final reconstructions of SVD-DIP are shown for the 1000 angle setting, along with the ground truth.

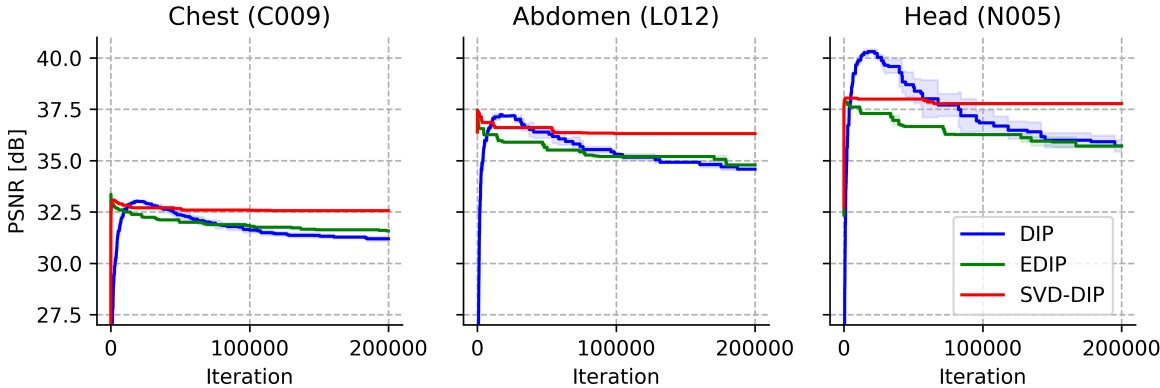

Figure 7: The mean and SD over 3 runs of the optimization of DIP, EDIP and SVD-DIP on samples from different body parts from LoDoPaB-Mayo, pretraining on LoDoPaB. The 1000 angle setting is used.

DIP achieves a higher max PSNR than EDIP and SVD-DIP. Recall that the architecture shown in Figure 1 is used for DIP, while for EDIP and SVD-DIP we employ the readily trained networks from (Baguer et al., 2020) with the architecture shown in Appendix C Figure 8(b). The latter, designed for a pure post-processing task, differs by being smaller, by including skip connections, and by *missing* a final sigmoid activation. The Head images have rather simplistic structures and can be fit particularly well by the architecture used for DIP (Figure 1), with the final sigmoid activation being an important inductive bias. In addition, the Head data is outside the training data distribution (LoDoPaB), while Chest and Abdomen are in or close to the training distribution. See Appendix C for additional

results with the respective other architecture choice. In all cases, DIP and EDIP show significant overfitting, which is effectively reduced with SVD-DIP.

## 5. Conclusion

In this work we built on the regularized fine-tuning of a pretrained DIP and adopted a novel strategy that restricts the learning to the adaptation of singular values of the unfolded network parameter tensor. We proposed a variant of the DIP, named SVD-DIP, that overall overcomes the need for early stopping, but sacrifices some speed relative to EDIP. This approach yields a more stable (less prone to overfit to noise) DIP optimization. The empirical results suggest that while the SVD-DIP reconstructive properties are usually on par with or slightly worse than those of DIP or EDIP, it loses very little in terms of PSNR, even after iterating for a long time.

## Acknowledgments

R.B. was supported by the i4health PhD studentship (UK EPSRC EP/S021930/1). J.L. was funded by the German Research Foundation (DFG; GRK 2224/1). The work of B.J. was partially supported by UK EPSRC grants EP/T000864/1 and EP/V026259/1. P.M. acknowledges support by DFG-NSFC project M-0187 of the Sino-German Center mobility programme.

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

# Appendix A. The Singular Value Decomposition is Sensible

While convolutions are linear and can thus be represented using a matrix for a fixed size input, these matrices are distinct from the input-size invariant matrices that we fold our tensors into. The former, for a tensor $W \in \mathbb{R}^{C_{out} \times C_{in} \times K \times K}$ are of the shape $\hat{W} \in \mathbb{R}^{C_{out}J^2 \times C_{in}J^2}$, for input images in $\mathbb{R}^{J \times J}$, while the folded matrix is of the form $W' \in \mathbb{R}^{C_{out} \times C_{in}K^2}$. For the former, it is direct that the SVD's use of matrix multiplication to compose the derived linear transformations yields the original convolution. For the latter, it may not be evident at first glance whether the multiplicative structure of the folded matrices is consistent with the composition of the convolutions they encode. Fortunately this is indeed the case. The key reason for this is that we unfold into $1 \times 1$ convolutions on the left. In the following, we demonstrate this rigorously:

For any integer $N$, we denote by the notation $[N]$ the set $\{1, \ldots, N\}$. Then for any integers $N$, $M$, (odd) $K$ and $J$, let

$$W := (w_{n,m,k_1,k_2})_{(n,m,k_1,k_2) \in [N] \times [M] \times [K]^2} \in \mathbb{R}^{N \times M \times K \times K},$$
$$X := (x_{m,j_1,j_2})_{(m,j_1,j_2) \in [M] \times [J]^2} \in \mathbb{R}^{M \times J \times J},$$

be two tensors. Then we can define the 2d convolution of $X$ with $W$ as a tensor

$$\text{conv2d}(W, X) := (y_{n,j_1,j_2})_{(n,j_1,j_2) \in [N] \times [J]^2} \in \mathbb{R}^{N \times J \times J}$$

with

$$y_{n,j_1,j_2} := \sum_{m=1}^{M} \sum_{k_1=1}^{K} \sum_{k_2=1}^{K} w_{n,m,k_1,k_2} \cdot X_{m,j_1+k_1-\frac{K-1}{2}-1,j_2+k_2-\frac{K-1}{2}-1}$$

with $X_{m,i_1,i_2} := 0$ when either of the indices $i_1, i_2$ is outside of $[J]$.

Then, for a given weight tensor $W \in \mathbb{R}^{N \times M \times K \times K}$ that can be represented by a matrix $A'B' = W' \in \mathbb{R}^{N \times MK^2}$, and a given feature map $X \in \mathbb{R}^{M \times J \times J}$, taking the 2-dimensional convolution of $X$ with respect to $W$ is identical to convolving $X$ by appropriate tensor representations of $B' \in \mathbb{R}^{R \times M \cdot K \cdot K}$ and (then) $A' \in \mathbb{R}^{N \times R \cdot 1 \cdot 1}$. Note that the second tensor only codes $1 \times 1$ convolutions. To prove this result, we need the following lemma:

**Lemma 1** *Let $M, K \in \mathbb{N}_{>0}$. Given a bijection between indices $f : [M] \times [K]^2 \to [MK^2]$, we can define a map that folds tensors into matrices:*

$$\Phi_f : \biguplus_{N=1}^{\infty} \mathbb{R}^{N \times M \times K \times K} \to \biguplus_{N=1}^{\infty} \mathbb{R}^{N \times MK^2}$$

$$(x_{n,m,k_1,k_2})_{(n,m,k_1,k_2) \in [N] \times [M] \times [K]^2} \mapsto (x_{n,f^{-1}(i)})_{(n,i) \in [N] \times [MK^2]}$$

*Let $g : [M] \times [K]^2 \to [MK^2]$ be a bijection, $N, R, J \in \mathbb{N}_{>0}$ be tensor dimensions, $X \in \mathbb{R}^{M \times J \times J}$ an input feature map. Let $W := (w_{n,m,k_1,k_2})_{(n,m,k_1,k_2) \in [N] \times [M] \times [K]^2} \in \mathbb{R}^{N \times M \times K \times K}$ be a tensor. Fix a bijection of indices $g : [M] \times [K]^2 \to [MK^2]$. Define $W' := \Phi_g^{-1}(W)$. Let $R \in \mathbb{N}_{>0}$, $A' \in \mathbb{R}^{N \times R}$, $B' \in \mathbb{R}^{R \times MK^2}$ such that $W' = A'B'$. Further, define $A :=$*

$\Phi^{-1}_{(Id_{[R]}, Id_{[1]}, Id_{[1]})^{-1}}(A')$, $B := \Phi_g^{-1}(B')$ *as the "unfolded" tensors of* $A', B'$. *Let* $X \in \mathbb{R}^{M \times J \times J}$ *be an arbitrary tensor/feature map. Then the following identity holds:*

$$conv2d(W, X) = conv2d(A, conv2d(B, X))$$

**Proof** We define the following tensors:

$$(a_{n,r,k_1,k_2})_{(n,r,k_1,k_2) \in [N] \times [R] \times [1] \times [1]} := A$$
$$(b_{r,m,k_1,k_2})_{(r,m,k_1,k_2) \in [R] \times [M] \times [K]^2} := B$$
$$(y_{r,j_1,j_2})_{(r,j_1,j_2) \in [R] \times [J]^2} := Y := conv2d(B, X)$$

Further, let $\pi_3, \pi_4$ be the mappings that extract the third and fourth entries of a given tuple. Then the proof reduces to the following elementary argument, exploiting the linearity of the operators involved:

$$(conv2d(A, conv2d(B, X)))_{n,j_1,j_2} = (conv2d(A, Y))_{r,j_1,j_2}$$

$$= \sum_{r=1}^{R} \sum_{k_1=1}^{1} \sum_{k_2=1}^{1} a_{n,r,k_1,k_2} \cdot Y_{r,j_1+k_1-\frac{1-1}{2}-1,j_2+k_2-\frac{1-1}{2}-1}$$

$$= \sum_{r=1}^{R} a_{n,r,1,1} \cdot Y_{r,j_1,j_2}$$

$$= \sum_{r=1}^{R} a_{n,r,1,1} \cdot \sum_{m=1}^{M} \sum_{k_1=1}^{K} \sum_{k_2=1}^{K} b_{r,m,k_1,k_2} \cdot X_{m,j_1+k_1-\frac{K-1}{2}-1,j_2+k_2-\frac{K-1}{2}-1}$$

$$= \sum_{r=1}^{R} a_{n,r} \cdot \sum_{i=1}^{MK^2} b_{r,i} \cdot X_{m,j_1+\pi_3(\Phi_g^{-1}(i))-\frac{K-1}{2}-1,j_2+\pi_3(\Phi_g^{-1}(i))-\frac{K-1}{2}-1}$$

$$= \sum_{i=1}^{MK^2} \sum_{r=1}^{R} a_{n,r} \cdot b_{r,i} \cdot X_{m,j_1+\pi_3(\Phi_g^{-1}(i))-\frac{K-1}{2}-1,j_2+\pi_3(\Phi_g^{-1}(i))-\frac{K-1}{2}-1}$$

$$= \sum_{i=1}^{MK^2} w_{n,i} \cdot X_{m,j_1+\pi_3(\Phi_g^{-1}(i))-\frac{K-1}{2}-1,j_2+\pi_3(\Phi_g^{-1}(i))-\frac{K-1}{2}-1}$$

$$= \sum_{m=1}^{M} \sum_{k=1}^{K} \sum_{k_2=1}^{K} w_{n,m,k_1,k_2} \cdot X_{m,j_1+k_1-\frac{K-1}{2}-1,j_2+k_2-\frac{K-1}{2}-1}$$

$$= (conv2d(W, X))_{n,j_1,j_2}.$$

This completes the proof of the claim and hence also the lemma. ∎

Note that in the statement, the embedding of $A'$ as a tensor does not permute its indices at all, only adding two more dimensions of size 1. Meanwhile, $B'$ is subjected to the same index transform $g^{-1}$ as the original matrix $W'$.

By applying the lemma twice, first for $U' \cdot (S'V')$, then for $U' \cdot V'$, we deduce that we can replace a convolutional layer with the three induced by the SVD decomposition.

## Appendix B. Experimental setup details

Python code is publicly available at https://github.com/educating-dip/svd_dip.

For the Lotus root dataset (Bubba et al., 2016), we use the forward operator matrix that has been provided with the dataset to implement fan-beam geometry. We use a TV reconstruction from all 120 angles as our ground truth reference, while our reconstruction task uses a sparse-view setting with 20 angles. The data is real-measured and thus already contains noise.

For the medical datasets, we use the ODL library (Adler et al., 2018) with the CUDA-accelerated ASTRA (van Aarle et al., 2015) backend to implement the parallel-beam geometries with 200 and 1000 angles, respectively. We simulate projections with (pre-log) Poisson noise, i.e., the full post-log model is given by

$$Ax + \nu = y, \qquad \nu = -Ax - \ln(N_1/N_0), \qquad N_1 \sim \mathrm{Pois}(N_0 \exp(-Ax)),$$

where $N_0 = 4096$ is the number of photons per detector pixel for an empty scan. Here, $x$ denotes the linear attenuation coefficients obtained from the Hounsfield unit values $x_{\mathrm{HU}}$ stored in DICOM files via $x = (20 - 0.02)x_{\mathrm{HU}}/1000 + 20$. After simulation, both images and projection data are divided by $\mu_{\max} = 81.35858$ to normalize images into the range $[0, 1]$. We use the appropriate Poisson regression loss that maximizes the likelihood under this model as our data discrepancy loss,

$$L_{\mathrm{Pois}}(A \cdot \mid y) = -\sum_{j=1}^{m} N_0 \exp(-y_j \mu_{\max})(-(A \cdot)_j \mu_{\max} + \ln(N_0)) - N_0 \exp(-(A \cdot)_j \mu_{\max}), \tag{2}$$

instead of the term $||A \cdot -y||_2^2$ in Equation 1 for all methods.

| Dataset to reconstruct | Noise | Data loss | $\gamma$ |
|---|---|---|---|
| Lotus root (Sparse 20) | real-measured | $\frac{1}{n}||A \cdot -y||_2^2$ | 1e-4 |
| LoDoPaB (Sparse 200): Chest | Poisson | $L_{\mathrm{Pois}}(A \cdot \mid y)$ | 4 |
| Mayo (Sparse 200): Chest, Abdomen, Head | Poisson | $L_{\mathrm{Pois}}(A \cdot \mid y)$ | 7 |
| Mayo (Sparse 1000): Chest, Abdomen, Head | Poisson | $L_{\mathrm{Pois}}(A \cdot \mid y)$ | 7 |

Table 2: Noise type, data losses and regularization parameters used for the different datasets. The respective data loss is used in place of $||A \cdot -y||_2^2$ in the objective Equation 1. The same choices are used for DIP, EDIP and SVD-DIP.

## Appendix C.  U-Net architecture choice and optimization hyperparameters

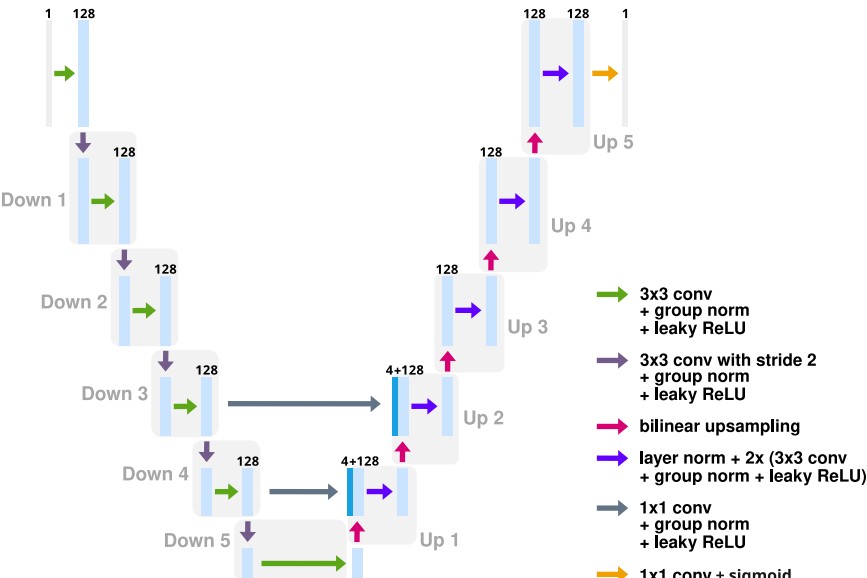

(*a*) U-Net used for all experiments on Ellipses-Lotus and Ellipses-LoDoPaB, as well as for non-pretrained DIP on LoDoPaB-Mayo.

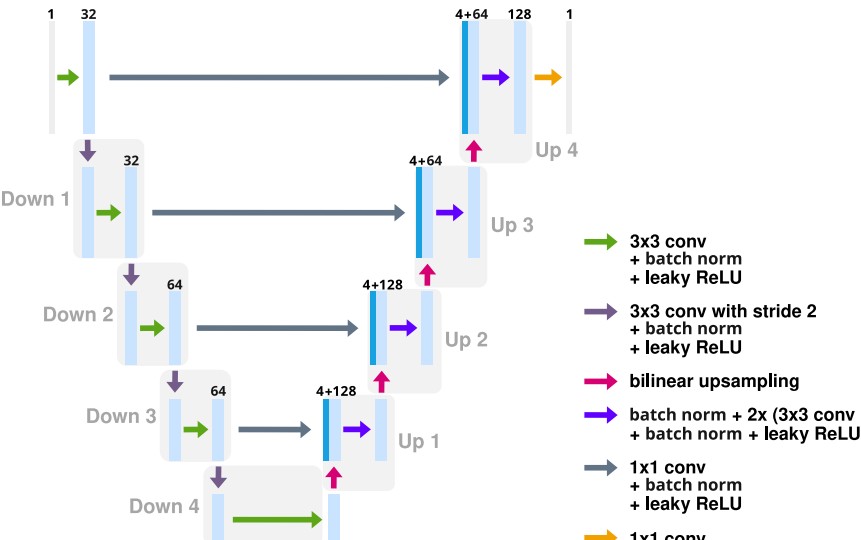

(*b*) U-Net used for EDIP and SVD-DIP on LoDoPaB-Mayo. Pretrained network weights were taken from (Baguer et al., 2020).

Figure 8: U-Net architectures

We use U-Net architectures for all our experiments, which are shown in Figure 8. We use Adam to optimize the objective Equation 1. Our default learning rate is 1e-4, and we apply a gradient clipping with maximum norm defaulted to 1.

We find that the gradient clipping maximum norm needs to be *well-adjusted* in order to be able to optimize a DIP when using an *output sigmoid activation*. This is the case for the non-pretrained DIP on LoDoPaB-Mayo, where we fine-tune the gradient clipping maximum norm to 1e-3. However, in all other results shown in the main text we use the default gradient clipping maximum norm of 1. For Ellipses-Lotus and Ellipses-LoDoPaB, we use the architecture shown in Figure 8(a), which is the same as the one used in (Baguer et al., 2020, Table B5) for LoDoPaB (except for the fact that we use group norm instead of batch norm layers).

For LoDoPaB-Mayo, we utilize the publicly available network parameters of FBP+U-Net on LoDoPaB/LoDoPaB-200 from (Baguer et al., 2020) to initialize EDIP and SVD-DIP using the architecture in Figure 8(b). We note that this architecture is sub-optimal for non-pretrained (vanilla) DIP reconstruction. We observe that different architectures can be favorable for DIP compared to EDIP and SVD-DIP, and vice versa. We use the FBP+U-Net architecture from (Baguer et al., 2020) (Figure 8(b)) for EDIP and SVD-DIP on LoDoPaB-Mayo, which we find to perform comparatively well on Chest and Abdomen, while for non-pretrained DIP on LoDoPaB-Mayo we use the architecture shown in Figure 8(a), based on the DIP architecture in (Baguer et al., 2020).

As an ablation study and to ensure a fair comparison between chosen architectures, we include additional investigations. We first pretrain on LoDoPaB with the architecture shown in Figure 8(a) for 40 epochs. Here we report our findings.

- Figure 9 and Figure 10 show results when using the architecture in Figure 8(b) for all methods, including the non-pretrained DIP. The non-pretrained DIP performs considerably worse in terms of max PSNR when compared to our main results in Figure 6 (left) and Figure 7, where the architecture in Figure 8(a) is used for the non-pretrained DIP.

- Figure 11 and Figure 12 show results when using the architecture in Figure 8(a) for all methods, including EDIP and SVD-DIP. While the final PSNR of EDIP and SVD-DIP here is better in many cases compared to our main results in Figure 6 and Figure 7, this advantage comes at the cost of highly fine-tuned optimization hyperparameters. Specifically, non-pretrained DIP worked best with a learning rate of 1e-4 and a gradient clipping maximum norm of 1e-3. For EDIP, we needed to lower the learning rate to 1e-5, again with a gradient clipping maximum norm of 1e-3. For SVD-DIP instead, the gradient clipping maximum norm needed to be set to 1, while using a learning of 1e-5 on Chest and Abdomen, and a learning rate of 1e-4 on Head. Part of the difficulty in tuning hyperparameters can be attributed to the sigmoid output activation, which we observe to require a well-tuned gradient clipping maximum norm. In Figure 11 and Figure 12, the lowered learning rate (1e-5) of EDIP naturally slows down its overfitting, so the rather high final PSNR values of EDIP are partially just an effect of slow optimization, thus not a clear indicator of stability; indeed, slow overfitting is still observed, in contrast to SVD-DIP, which remains stable. We also note that the max PSNR of EDIP for Chest and Abdomen in Figure 11 (left) is on par when

compared to our main results in Figure 6 (left), so unlike the non-pretrained DIP, in these cases EDIP does not benefit from the architecture in Figure 8(a) in terms of max PSNR.

To summarize our findings: the standard DIP usually benefits from overparameterization, omitting skip connections at high-resolution scales, and from a final sigmoid activation (to a different degree depending on the image class). Pretrained DIP (EDIP and SVD-DIP) can leverage skip connections, and while a final sigmoid activation can also improve EDIP and SVD-DIP, it needs a careful fine-tuning of the optimization hyperparameters.

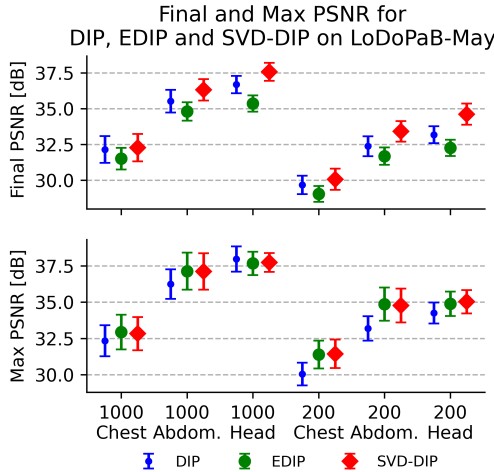

Figure 9: This figure is like Figure 6 (left), except that DIP here also uses the architecture shown in Figure 8(b).

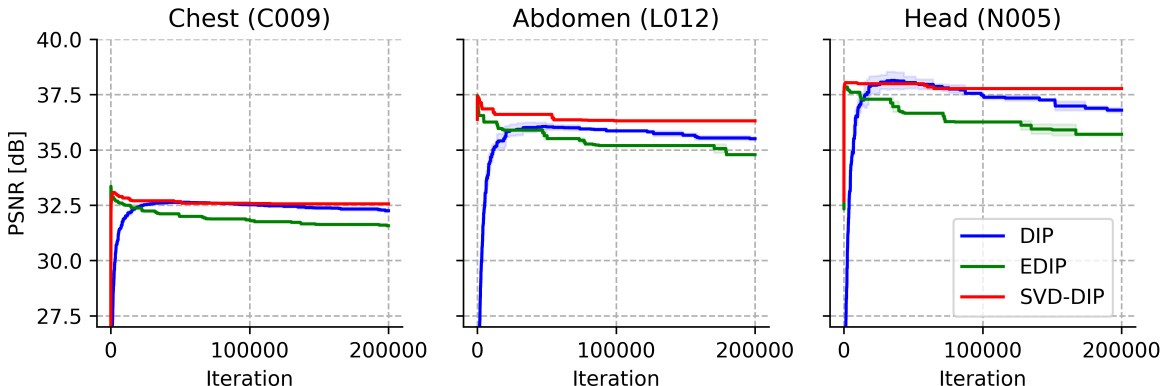

Figure 10: This figure is like Figure 7, except that DIP here also uses the architecture shown in Figure 8(b).

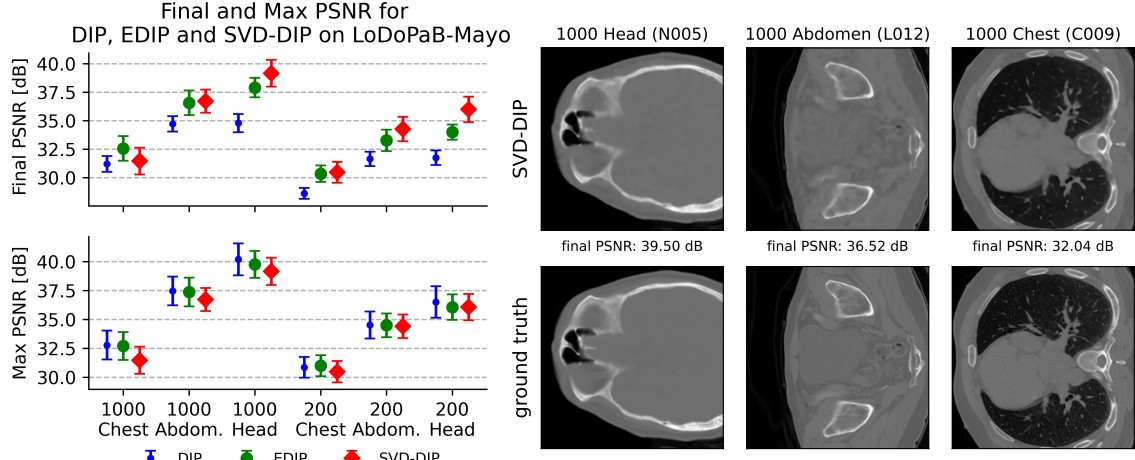

Figure 11: This figure is like Figure 6, except that EDIP and SVD-DIP here also use the architecture shown in Figure 8(a).

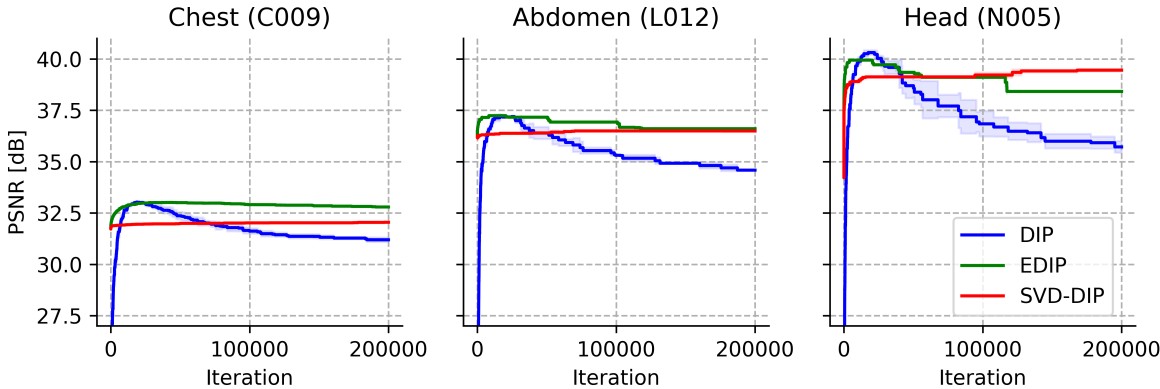

Figure 12: This Figure is like Figure 7, except that EDIP and SVD-DIP here also use the architecture shown in Figure 8(a).

## Appendix D. Additional results

Table 3: PSNR values for Ellipses-Lotus (Sparse 20), 200000 Iterations.

| Method | Final | Max (Iteration) | Max−Final | Final−Init |
|---|---|---|---|---|
| DIP | 31.07 | 31.69 (5878) | 0.6166 | 22.12 |
| EDIP | 31.09 | 31.95 (1289) | 0.8541 | 4.11 |
| SVD-DIP (no reduction) | 31.60 | 31.60 (198047) | 0.0000 | 4.62 |
| SVD-DIP (50% rd.) | 31.64 | 31.64 (189325) | 0.0063 | 5.07 |
| SVD-DIP (10% thresh.) | 31.58 | 31.58 (195288) | 0.0048 | 4.69 |
| SVD-DIP (no pretraining, no red.) | 29.38 | 29.38 (198239) | 0.0000 | 20.44 |

Table 4: PSNR values for Ellipses-LoDoPaB (Sparse 200) samples, 50000 Iterations.

| | DIP | | Pretraining | EDIP | | SVD-DIP | |
|---|---|---|---|---|---|---|---|
| Test Set Index | Final | Max | Init | Final | Max | Final | Max |
| 6 | 34.81 | 37.77 | 32.6 | 35.07 | 37.62 | 37.24 | 37.28 |
| 9 | 30.67 | 34.59 | 30.22 | 30.98 | 34.42 | 33.87 | 33.93 |
| 11 | 29.69 | 33.93 | 28.65 | 30.55 | 33.6 | 33.05 | 33.11 |
| 13 | 36.17 | 39.74 | 35.57 | 36.97 | 39.4 | 38.86 | 38.91 |
| 17 | 31.63 | 36.39 | 31.12 | 31.81 | 35.85 | 35.29 | 35.45 |
| 19 | 31.33 | 34.05 | 30.48 | 31.47 | 33.77 | 33.52 | 33.71 |
| 22 | 30.56 | 33.46 | 31.14 | 30.97 | 33.33 | 32.86 | 33.04 |
| 29 | 32.76 | 36.01 | 31.43 | 32.94 | 35.87 | 35.27 | 35.42 |
| 34 | 31.47 | 33.06 | 24.76 | 31.4 | 32.84 | 32.73 | 32.85 |
| 43 | 31.75 | 34.42 | 29.13 | 31.7 | 34.2 | 33.78 | 33.91 |
| Mean | 32.08 | 35.34 | 30.51 | 32.39 | 35.09 | 34.65 | 34.76 |

Table 5: PSNR values for LoDoPaB-Mayo (Full 1000), Chest/C samples, 40000 iterations.

| | DIP | | Pretraining | EDIP | | SVD-DIP | |
|---|---|---|---|---|---|---|---|
| Patient | Final | Max | Init | Final | Max | Final | Max |
| C001 | 31.85 | 34.55 | 23.31 | 32.30 | 34.96 | 33.53 | 34.60 |
| C002 | 29.95 | 30.83 | 24.44 | 30.23 | 31.11 | 30.75 | 31.03 |
| C004 | 31.34 | 32.44 | 24.01 | 31.49 | 32.57 | 32.07 | 32.42 |
| C009 | 31.19 | 33.05 | 20.86 | 31.57 | 33.36 | 32.56 | 33.08 |
| C012 | 30.49 | 31.75 | 25.96 | 30.67 | 32.20 | 31.41 | 32.15 |
| C016 | 31.43 | 32.53 | 25.58 | 31.61 | 32.69 | 32.17 | 32.58 |
| C019 | 32.20 | 34.63 | 22.11 | 32.66 | 34.50 | 33.84 | 34.54 |
| C021 | 31.69 | 32.94 | 19.77 | 32.05 | 32.48 | 32.25 | 32.54 |
| C023 | 30.27 | 31.18 | 23.22 | 30.56 | 31.47 | 31.14 | 31.49 |
| C027 | 31.55 | 33.91 | 24.48 | 31.92 | 34.01 | 33.01 | 33.89 |
| Mean | 31.20 | 32.78 | 23.38 | 31.51 | 32.93 | 32.27 | 32.83 |

Table 6: PSNR values for LoDoPaB-Mayo (Sparse 200), Chest/C samples, 100000 Iterations.

| | DIP | | Pretraining | EDIP | | SVD-DIP | |
|---|---|---|---|---|---|---|---|
| Patient | Final | Max | Init | Final | Max | Final | Max |
| C001 | 28.95 | 31.94 | 32.29 | 29.64 | 33.04 | 30.93 | 33.09 |
| C002 | 27.83 | 29.32 | 29.71 | 28.09 | 30.01 | 28.94 | 30.10 |
| C004 | 28.81 | 30.93 | 28.06 | 29.16 | 31.22 | 29.97 | 31.11 |
| C009 | 28.55 | 30.90 | 30.17 | 29.08 | 31.63 | 30.27 | 31.76 |
| C012 | 28.00 | 29.94 | 29.82 | 28.33 | 30.67 | 29.34 | 30.85 |
| C016 | 28.72 | 30.65 | 28.02 | 29.06 | 30.96 | 29.83 | 31.02 |
| C019 | 29.54 | 32.38 | 27.07 | 30.03 | 32.55 | 31.54 | 32.79 |
| C021 | 28.90 | 30.96 | 16.45 | 29.24 | 31.12 | 29.94 | 30.85 |
| C023 | 28.11 | 29.92 | 24.39 | 28.54 | 30.30 | 29.41 | 30.36 |
| C027 | 28.75 | 31.60 | 31.52 | 29.24 | 32.45 | 30.51 | 32.46 |
| Mean | 28.62 | 30.85 | 27.75 | 29.04 | 31.39 | 30.07 | 31.44 |

Table 7: PSNR values for LoDoPaB-Mayo (Full 1000), Abdomen/L samples, 200000 Iterations.

|  | DIP | | Pretraining | EDIP | | SVD-DIP | |
| --- | --- | --- | --- | --- | --- | --- | --- |
| Patient | Final | Max | Init | Final | Max | Final | Max |
| L004 | 35.05 | 37.74 | 32.04 | 35.12 | 37.56 | 36.80 | 37.37 |
| L006 | 35.68 | 39.49 | 35.32 | 35.44 | 38.70 | 37.44 | 38.81 |
| L012 | 34.58 | 37.26 | 36.40 | 34.79 | 37.46 | 36.31 | 37.42 |
| L014 | 35.51 | 39.32 | 36.20 | 35.42 | 39.09 | 37.12 | 38.91 |
| L019 | 34.70 | 37.59 | 36.10 | 34.16 | 37.94 | 36.11 | 38.09 |
| L024 | 34.98 | 37.86 | 22.22 | 35.43 | 36.55 | 36.63 | 36.63 |
| L027 | 34.12 | 36.56 | 27.16 | 34.81 | 35.87 | 35.93 | 35.93 |
| L030 | 33.77 | 35.45 | 23.51 | 33.83 | 35.02 | 34.97 | 35.01 |
| L033 | 33.58 | 35.87 | 32.22 | 33.67 | 35.56 | 35.21 | 35.60 |
| L035 | 35.18 | 37.45 | 32.23 | 35.33 | 37.54 | 36.65 | 37.33 |
| Mean | 34.72 | 37.46 | 31.34 | 34.80 | 37.13 | 36.32 | 37.11 |

Table 8: PSNR values for LoDoPaB-Mayo (Sparse 200), Abdomen/L samples, 100000 Iterations.

|  | DIP | | Pretraining | EDIP | | SVD-DIP | |
| --- | --- | --- | --- | --- | --- | --- | --- |
| Patient | Final | Max | Init | Final | Max | Final | Max |
| L004 | 31.87 | 35.30 | 34.39 | 32.12 | 35.25 | 34.25 | 35.52 |
| L006 | 32.38 | 35.98 | 33.39 | 32.12 | 36.36 | 33.96 | 36.19 |
| L012 | 31.62 | 34.34 | 34.24 | 31.69 | 35.31 | 33.35 | 35.04 |
| L014 | 32.40 | 36.43 | 33.85 | 31.99 | 36.50 | 33.97 | 36.45 |
| L019 | 31.02 | 34.69 | 33.83 | 30.92 | 35.26 | 33.01 | 35.35 |
| L024 | 32.02 | 34.72 | 28.32 | 32.37 | 34.09 | 33.92 | 33.97 |
| L027 | 31.17 | 33.33 | 30.56 | 31.63 | 33.78 | 32.87 | 33.77 |
| L030 | 30.94 | 32.82 | 31.56 | 31.03 | 33.26 | 32.51 | 33.24 |
| L033 | 30.65 | 32.83 | 32.32 | 30.58 | 33.17 | 32.10 | 32.90 |
| L035 | 32.39 | 34.73 | 34.80 | 32.37 | 35.61 | 34.17 | 35.27 |
| Mean | 31.65 | 34.52 | 32.72 | 31.68 | 34.86 | 33.41 | 34.77 |

Table 9: PSNR values for LoDoPaB-Mayo (Full 1000), Head/N samples, 200000 Iterations.

| | DIP | | Pretraining | EDIP | | SVD-DIP | |
|---|---|---|---|---|---|---|---|
| Patient | Final | Max | Init | Final | Max | Final | Max |
| N001 | 35.76 | 42.38 | 28.69 | 35.99 | 39.26 | 38.57 | 38.81 |
| N003 | 33.54 | 37.54 | 17.33 | 34.69 | 36.55 | 36.67 | 36.68 |
| N005 | 35.71 | 40.31 | 32.53 | 35.71 | 37.88 | 37.77 | 38.05 |
| N012 | 35.06 | 38.92 | 30.50 | 34.86 | 37.40 | 37.00 | 37.38 |
| N017 | 34.15 | 38.65 | 16.71 | 34.73 | 36.66 | 36.96 | 36.96 |
| N021 | 34.68 | 40.93 | 19.35 | 35.78 | 38.07 | 38.04 | 38.04 |
| N024 | 33.54 | 39.91 | 24.69 | 34.47 | 36.75 | 36.74 | 36.98 |
| N025 | 34.65 | 41.07 | 26.50 | 35.53 | 37.77 | 37.89 | 38.04 |
| N029 | 35.02 | 41.29 | 23.43 | 35.73 | 38.25 | 37.88 | 38.16 |
| N030 | 35.71 | 40.97 | 28.94 | 36.05 | 38.11 | 38.11 | 38.20 |
| Mean | 34.78 | 40.20 | 24.87 | 35.35 | 37.67 | 37.56 | 37.73 |

Table 10: PSNR values for LoDoPaB-Mayo (Sparse 200), Head/N samples, 100000 Iterations.

| | DIP | | Pretraining | EDIP | | SVD-DIP | |
|---|---|---|---|---|---|---|---|
| Patient | Final | Max | Init | Final | Max | Final | Max |
| N001 | 32.72 | 38.75 | 28.06 | 32.72 | 36.09 | 35.61 | 36.31 |
| N003 | 30.79 | 33.82 | 19.90 | 31.34 | 33.04 | 33.49 | 33.56 |
| N005 | 32.55 | 36.99 | 29.94 | 32.43 | 35.21 | 34.48 | 35.13 |
| N012 | 31.64 | 35.00 | 27.91 | 31.84 | 34.42 | 33.72 | 34.43 |
| N017 | 31.25 | 35.55 | 21.17 | 32.21 | 34.30 | 34.30 | 34.36 |
| N021 | 31.96 | 37.58 | 22.86 | 32.77 | 35.39 | 35.77 | 35.84 |
| N024 | 30.80 | 36.08 | 22.82 | 31.18 | 34.23 | 33.90 | 34.31 |
| N025 | 31.55 | 36.78 | 25.26 | 32.62 | 35.34 | 34.67 | 35.16 |
| N029 | 32.09 | 37.69 | 25.37 | 32.73 | 35.51 | 34.97 | 35.62 |
| N030 | 32.12 | 36.78 | 27.91 | 32.77 | 35.34 | 35.18 | 35.51 |
| Mean | 31.75 | 36.50 | 25.12 | 32.26 | 34.89 | 34.61 | 35.02 |

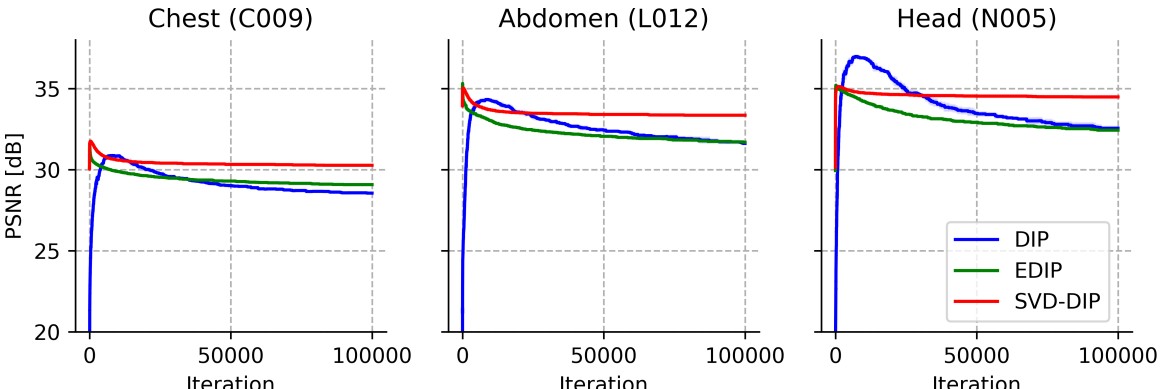

Figure 13:  The mean and SD over 3 runs of the optimization of DIP, EDIP and SVD-DIP on LoDoPaB-Mayo (Sparse 200) for samples C009, L012 and N005. Each line represents the mean over 3 runs.

## Appendix E.  Using TV for Early Stopping

Simple strategies for early stopping (ES) are hard to come by for DIP or its variants; there is extensive literature on the subject. Most ES approaches are tailored to either a specific problem or model, and often do not generalize to different settings.  For example, the interesting approach proposed by (Shi et al., 2021) requires a specialized architecture, while (Jo et al., 2021) suggest a method for ES in the context of denoising based on Stein unbiased risk estimator (SURE). First we briefly investigated whether the prior (or regulariser) could act as a proxy for an ES criterion, and discuss SURE later.  Specifically we investigated whether it is possible to use the TV for ES. From Figure 14, the TV fails to yield no useful information about the PSNR curve. The TV trends upwards, with no discernible difference in this trend when the PSNR falls, making it a poor tool for predicting when overfitting occurs.

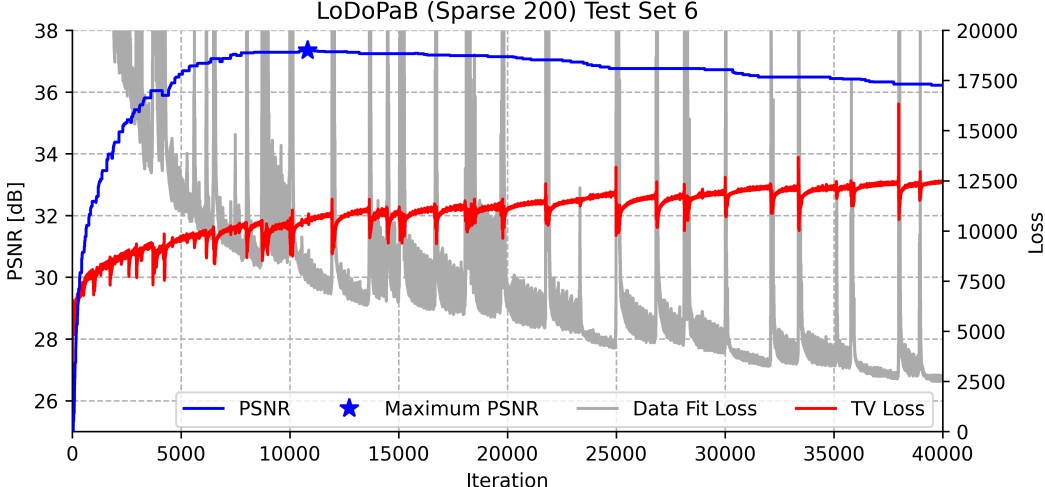

Figure 14: PSNR and mean loss output TV of DIP on LoDoPaB.

## Appendix F. Is SURE a viable Method for CT?

One naturally may ask whether a DIP which utilizes Stein's unbiased risk estimator (SURE) could serve as a better baseline than the DIP we have used. Indeed, SURE has reported state-of-the-art results on several image-restoration tasks such as de-noising, de-blurring, and super-resolution. However, we argue that its applicability to more "more ill-posed" inverse problems in imaging is currently an open field of research. Recall that the SURE is an unbiased estimate of the mean square error (MSE). Clearly, the MSE has an explicit dependence on the unknown ground truth image. In several settings, there have been very successful ways to estimate the MSE. Initially proposed by Stein for the independent, identically distributed (i.i.d.) Gaussian model, SURE has been extended to exponential distributions (GSURE) by (Eldar, 2009). Lately, GSURE has been successfully applied in many works (Metzler et al., 2018; Abu-Hussein et al., 2021; Jo et al., 2021), which have demonstrated excellent performance of the method on de-noising, de-blurring, super-resolution or, compressed sensing tasks.

In the presence of an operator, the use of SURE is less direct than for de-noising. For example, in a Gaussian linear model (e.g., $y = Ax + \nu$), the projected GSURE provides an unbiased estimate of the projected MSE, which is the expected error of the projections in the column space of the operator $A$. Hence, when the matrix $A$ is rank deficient, SURE is a sub-optimal estimator as it does not estimate the orthogonal complement. Indeed, for very ill-posed inverse problems, e.g., in the highly under-sampled setting (i.e., rank-deficient setting), (Metzler et al., 2018) show that the GSURE-based projected MSE is a poor approximation of the actual MSE. To address this issue, (Aggarwal et al., 2022) propose ENSURE, which generalizes the classical SURE and GSURE formulation where the images are sampled by different operators $A$, chosen randomly from a set. Unfortunately, ENSURE is not applicable to CT problems in conjunction to the unsupervised DIP framework.

To confirm these observations, we run the following set of experiments. Figure 15 shows that DIP-SURE significantly outperforms DIP when optimized with the least-square

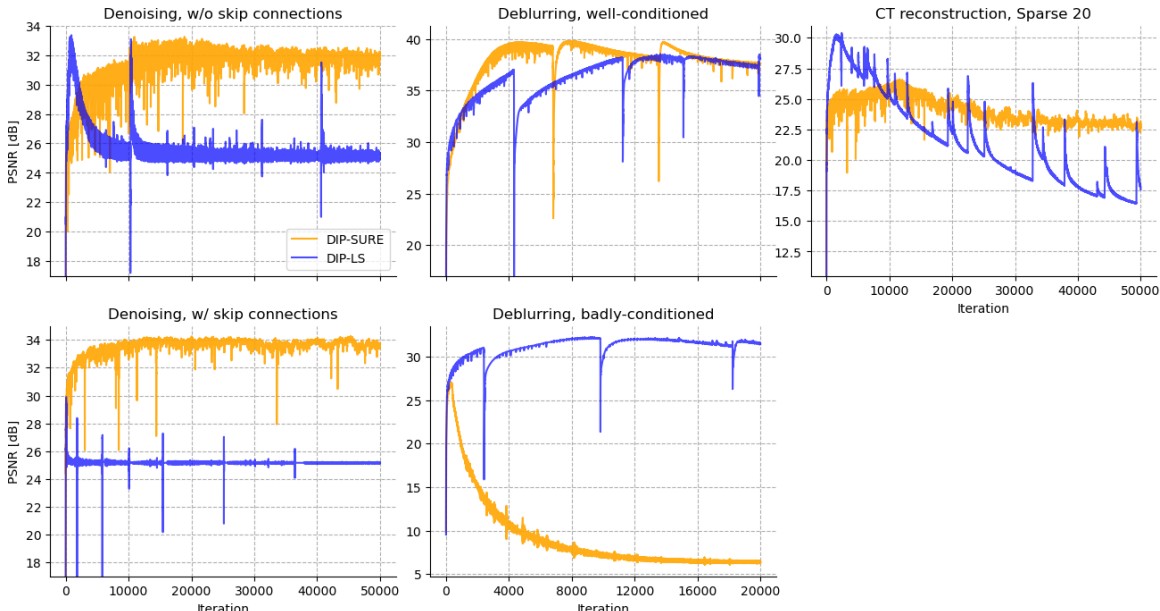

Figure 15: DIP with LS loss compared to DIP with SURE loss for different tasks on the Lotus root. For denoising (first column), we tested the network both without and with skip connections. For deblurring (second column), a well-conditioned ($\kappa(A) \sim 14$) and a badly-conditioned ($\kappa(A) \sim 1335$) problem were tested, both using skip connections in the network. For CT reconstruction (right), we use the sparse 20-angle setting like in the main experiments (but without TV regularization for both methods).

objective (DIP-LS) for the de-noising task, where we use the lotus ground truth image $x$ and simulate the noisy image $y$ by adding white noise $\nu \sim \mathcal{N}(0, \sigma^2 I)$ to $x$, with $\sigma$ being half of the mean of $|x|$. The difference in the performance between DIP-SURE and DIP-LS is further accentuated if the architecture uses skips connections at every depth. Note that in the above works on SURE, all use a U-Net architecture equipped with skip connections at every depth since skip connections boost the network capacity to overfit to noise.

We then compare DIP-SURE vs. DIP-LS on Gaussian deblurring tasks. We construct two matrices $A$ using the same Gaussian kernel but applying different Tikhonov regularization, thus adding either to the diagonal entries the mean of all diagonal entries, or 0.01 percent of the diagonal mean. This results in two blurring matrices with different condition numbers (i.e., $\kappa(A) \sim 14$ and $\kappa(A) \sim 1335$ respectively), i.e. different degree of ill-posedness. DIP-SURE outperforms DIP-LS in the mildly ill-conditioned case, whilst, as we would expect, fails on the more ill-posed task. Similarly, on the Lotus reconstruction tasks, due to ill-posedness of the reconstructive task, DIP-SURE fails to match DIP-LS. In CT reconstruction, the operator usually has very small singular values and for sparse-view scans it is also rank-deficient, rendering the problem ill-posed (Buzug, 2011). In sum, our investigation suggests that SURE is not a viable option for our setting.

