# OpenReview forum: "SVD-DIP: Overcoming the Overfitting Problem in DIP-based CT Reconstruction"
_MIDL.io/2023/Conference — MIDL 2023 Poster_

### Official Review · Reviewer_AhP7 · 2023-02-01

**Confidence:** 3
**Preliminary Rating:** 3
**Recommendation:** Poster

**Summary:**

The paper proposes to combine singular value fine-tuning with the educated deep image prior, to improve the basic deep image prior and alleviate the overfitting problem. In particular they only tune the singular values of the SVD decomposed weight matrices from a previously pretrained network in context of the deep image prior. The authors claim that this reduction in tune-able parameters makes the network less prone to overfit on the actual image.

**Strengths:**

+ The paper is clearly written and easy to understand
+ I personally find the idea quite cool and really fitting for this use case
+ The experiment setup seems convincing
+ The performance on most datasets is convincing

**Weaknesses:**

While it brings together to ideas from two different fields, it's probably not the most novel method (but I personally do not see this as a weakness in this case)

However, while I really liked the paper, there are 3 points, that I think could be addressed to make the paper even stronger:
- The authors claim, that one major flaw of DIP is that it needs to be retrained each time and is thus quite time consuming. Looking at most of the results, it seems that SVD-DIP is even more time consuming (since the fine-tuning needs way more time to reach the max performance).
- The overall achievable max performance of SVD-DIP seems to suffer in some cases, due to the constrained parameter space. I feel like this should be discussed in more detail.
- Did the authors accidentally confuse SVD-DIP and DIP for the Mayo experiments ? (please double check, also for the suppl. Figure ). If yes, every thing would make sense and this point can be omitted. Otherwise the results here contradict the previous experiments and most claims  the authors make during the paper. Here, SVD-DIP seems to be more prone to overfitting and have the training dynamics of the DIP model from the experiments before, while DIP seems less prone to overfitting. While SVD-DIP shows the best max performance in some cases, it overall undermines the added value SVD-DIP can bring. So if the results hold up, I would find it hard to make an actual claim for SVD-DIP (compared with DIP) at least regarding the overfitting capabilities (I feel like it is still an interesting method, but kind of the main point made in this paper falls apart).

**Deanonymize Review:**

no

**Detailed Comments:**

If you give the mean over 3 runs, why not plot the std/ value range in all plots similar to Fig.3&4 in all cases ?

**Paper Type:**

methodological development

**Questions To Address In The Rebuttal:**

Please answer especially the points regarding the mayo experiments (why the training dynamics appear to change for that dataset and if SVD-DIP still has some merit in this case ).
And if possible discuss the performance differences and time requirements.

---

### Official Review · Reviewer_wSQu · 2023-02-03

**Confidence:** 5
**Preliminary Rating:** 2

**Summary:**

The article proposes to use Singular Value fine-tuning methodology in the
    warm-start approach used for DIPs. First a model is trained on some data,
    SVDs are computed for weights between different layers, then only the
    singular values are fine-tuned for fitting the DIP model to a new image. Due
    to the constraint in the weight updates, the model does not overfit. This
    behavior is demonstrated in multiple experiments on CT reconstruction.

**Strengths:**

1. Stabilizing DIP training is a very relevant topic for medical image
       analysis.
2. Combining two recent techniques to leverage their advantages is a good
       approach. The investigated approach definitely makes sense.
3. It is intriguing to see that pre-training on synthetic data can lead to
       accurate reconstructions with fine-tuning. That surely increases the
       applicability of such methods.
4. The experiments are diverse. They worked on micro-CT, synthetic data, and
       real medical images in this work.

**Weaknesses:**

1. More information on the reconstruction problem would help readers.
2. Comparison with Stein's unbiased risk estimator (SURE) is
       crucial. Omission of such comparison reduces the quality of the article
       substantially.
3. Singular value fine-tuning is a very strong regularization, and it has
       some costs. Figure 5 and Table 1 show this clearly. DIP methods can
       achieve higher values when trained without a regularization than the
       proposed technique. Therefore, methods that can stabilize the training,
       e.g., SURE, has the potential to outperform the proposed technique. A
       careful analysis of the costs of the strong regularization need to be
       studied.
4. The only metric used here is PSNR. How do authors compute PSNR? This may
       not be enough. I strong recommend using a metric that assesses fidelity,
       e.g., NMSE.

**Deanonymize Review:**

no

**Paper Type:**

both

**Questions To Address In The Rebuttal:**

1. Please compare with SURE.
2. Please report fidelity metrics.
3. Please comment on the costs of the strong regularization.
4. Please provide further details on the experimental setup, especially on
       details pertinent to the reconstruction problem.

---

### Official Review · Reviewer_suEt · 2023-02-04

**Confidence:** 3
**Preliminary Rating:** 4

**Summary:**

This paper describes a strategy to prevent overfitting in the Deep Image Prior. In particular, the paper proposes to pretrain the weights of the DIP, compute an SVD of each of the weight matrices, and only tune the singular values of the resulting SVD. The paper reports experimental results on a CT reconstruction task in a lotus root as well as the LoDoPaB and Mayo datasets.

**Strengths:**

- The paper is clearly written and the method is well-explained. The description of prior work is well done and frames the current work well relative to previous advances.
- The method shows promise on the lotus root data, where it stably achieves the peak PSNR achieved by EDIP before it begins to overfit.
- While the idea of using an SVD of the weight matrix and training only the singular values has been proposed before (Sun et al., as cited in the paper), the idea of applying this to DIP is novel.

**Weaknesses:**

- The paper is missing comparisons to important baselines (DIP with standard regularization techniques).
- While the experimental results are convincing on the lotus root dataset, the advantages of the method are less clear on the other two transfer tasks.
- The paper does not summarize trends in the results across the entire dataset, largely focusing on results on individual examples.

**Deanonymize Review:**

no

**Detailed Comments:**

- As detailed in Eq. 1, it is standard to use a regularizer (e.g. TV regularization) to prevent overfitting with the DIP. In addition to the “vanilla” DIP baseline shown in this paper, a comparison should be made to DIP with the TV regularizer, to understand how the gains from the proposed method compare. If the claim is that TV regularization could be used in addition to the proposed SVD approach, then an additional experiment is needed to show both SVD-DIP+TV and how much that improves over DIP+TV.
- On the LoDoPaB dataset (Fig 5), the method does not achieve the peak SNR achieved by standard DIP or EDIP. While it does achieve a higher stable value, it is possible that simple tests could be used to perform early stopping on the DIP approach and yield a higher quality reconstruction. For example, early stopping could be performed by monitoring the total variation metric of the reconstruction at each iteration. Such a baseline should be reported. Further, on the Mayo dataset (Fig 7), the final value obtained by SVD-DIP is not significantly better than that obtained by standard DIP, which does seem to be relatively stable — in this case, there is not a clear benefit from the SVD-DIP method. Overall, it is not clear that the method robustly provides an improvement over standard E/DIP.
- As far as I can tell, most of the results in the main paper are reported on individual samples (all of the PSNR curves, which should also plot the variance over the three runs). The appendix tables contain results on more individual examples, but do not appear to contain all of them (e.g. in Table 3: only samples 6, 9, 11, etc. are included). The main paper should contain a table and/or figure that summarizes the statistics across all examples in each of the datasets, instead of only showing results on select examples. I can’t tell if the results in Figure 6 are summarized over a dataset or for individual examples; they should be over the entire dataset and should also show a variance measure to give a sense for variability in the results.
- Figure 6 may be clearer/easier to interpret if the improvement over a final DIP value or algorithmically selected peak DIP value (e.g. with early stopping) is plotted. It is hard to interpret a low “Max - Final” value, for example, if both the Max and Final PSNR are low. I would recommend showing the improvement/change over DIP in the main paper, and then pointing to the more detailed metrics in the Appendix.
- The statement of Lemma 1 is extremely difficult to follow. I can follow the proof below and its connection to the statement that convolution with W is equivalent to convolution with a tensor representation of B’ followed by a tensor representation of A’, but the lemma statement itself is confusing. If there is a way to simplify the notation of the lemma, that would go a long way in making this section more readable. For example, it may be helpful to simplify the lemma to involve only a statement of the form $conv2d(W,X) = conv2d(A, conv2d(B,X))$, which would avoid the complicated notation and commutative diagram in the lemma. It is very possible I am missing a subtle point about the proof/statement here; if so, please feel free to clarify (and in general, to add more explanatory text to the proof).

**Paper Type:**

methodological development

**Questions To Address In The Rebuttal:**

- How does the proposed method compare to DIP with a regularizer (e.g. TV) and with early stopping based on monitoring a TV-like quantity?
- How much do the reported results vary across all samples in the entire dataset? (See above)
- What exactly is being stated in Lemma 1 in the Appendix and is there a way to simplify the notation to make this clearer? (See above)

---

### Meta-Review · Area_Chair_Uj97 · 2023-02-26

**Recommendation:** Accept (Poster)
**Confidence:** 3

**Metareview:**

This paper presents a method to deal with overfitting in DIP-based reconstruction. The paper receives thorough and thoughtful reviews that seem to have understood the paper and pointed out a few serious issues.

The most important issue was the lack of comparison to SURE, which is the crucial fundamental comparison here. The fact that this experiment was not done originally is quite concerning, and led to negative ratings from two of the three reviewers. The authors used the rebuttal period to do a deep dive in this direction, both including detailed answers and experiments with SURE-based frameworks. Overall, one reviewer was satisfied, while the other did not answer. I appreciate the authors' efforts here. I am, however, quite puzzled as to why this was not present in the original submission, as it seems central. While the authors' argument (as far as I can tell) that the current papers using SURE are likely suboptimal for their setting might(?) be reasonable, omitting a good-faith comparison based on this does not seem right -- indeed the experiment is necessary to validate the hypothesis/argument. It will also make the paper complete.

Nevertheless, the experiment is now done, and the paper is stronger. As the reviewers' note, the improvements of the current paper are overall modest. To me, the updated paper with the SURE experiments just pass the threshold for acceptance. Is is quite a borderline paper, and the fact that such an important fundamental comparison was missed originally lowers my opinion, as it makes it more challenging to assess the insights the authors were aiming for (e.g. if the SURE comparison was there originally, an initial review may well have been able to assess and give feedback on a much more complete, almost different, paper).